# Influence of Age and Mating Status on Pheromone Production in a Powderpost Beetle *Lyctus africanus* (Coleoptera: Lyctinae)

**DOI:** 10.3390/insects12010008

**Published:** 2020-12-25

**Authors:** Titik Kartika, Nobuhiro Shimizu, Setiawan Khoirul Himmi, Ikhsan Guswenrivo, Didi Tarmadi, Sulaeman Yusuf, Tsuyoshi Yoshimura

**Affiliations:** 1Research Center for Biomaterials, Indonesian Institute of Sciences, Bogor 16911, Indonesia; khoirul_himmi@biomaterial.lipi.go.id (S.K.H.); ikhsan.guswenrivo@biomaterial.lipi.go.id (I.G.); didi@biomaterial.lipi.go.id (D.T.); sulaeman@biomaterial.lipi.go.id (S.Y.); 2Faculty of Bioenvironmental Science, Kyoto University of Advanced Science, Kyoto 621-8555, Japan; shimizu.nobuhiro@kuas.ac.jp; 3Research Institute for Sustainable Humanosphere, Kyoto University, Kyoto 611-0011, Japan; tsuyoshi@rish.kyoto-u.ac.jp

**Keywords:** pheromone, *Lyctus africanus*, powderpost beetle, age, mating

## Abstract

**Simple Summary:**

Powderpost beetles, such as *Lyctus africanus*, are a common pest group for dried cured wood. The damage is slow and inconspicuous; thus, the infestation is mostly identified belatedly due to a lack of knowledge of how to locate and monitor it. *L. africanus* produces a pheromone, a chemical compound to attract other beetles. This pheromone has been determined and suggested as a monitoring tool for *L. africanus*. Here, we examined the physiological and behavioral parameters that affect pheromone production. We found that food availability may affect pheromone production in adult *L. africanus* males. In addition, of three components in male *L. africanus* aggregation pheromones, major compounds **2** (3-pentyl dodecanoate) and **3** (3-pentyl tetradecanoate) may be affected by age, not mating status, while compound **1** (2-propyl dodecanoate) was produced steadily and was affected by mating status. This suggests compounds **2** and **3** might have an important function in aggregation behavior, especially in signaling for mating opportunities. We also were able to clarify the minor effect of compound **1** in the aggregation pheromone of *L. africanus*, although not its role. The present information will be helpful in understanding the chemical communication of these insects, which may be important for the development of improved pheromone-based management strategies for controlling *Lyctus* beetles.

**Abstract:**

Powderpost beetles such as *Lyctus africanus* are a common pest group for dried cured wood, causing significant harm to wood and wood products. We examined the life span and effects of aging and mating status on pheromone production in the powderpost beetle *L. africanus* (Coleoptera: Lyctinae). Experiments compared starved and unstarved male groups, and chemical analysis was used to determine factors affecting pheromone production. Regarding lifespan, male beetles provided food survived up to 14 weeks, while starved beetles died before the fifth week. Thus, an adult *L. africanus* male may require food throughout its lifespan, and food availability may affect pheromone production. There was no significant difference in the quantity of two major pheromone compounds, compound **2** (3-pentyl dodecanoate) and **3** (3-pentyl tetradecanoate) between mated and un-mated males. On the other hand, a minor compound, compound **1** (2-propyl dodecanoate) showed increased quantity after mating. The two major compounds were produced in low amounts by young *L. africanus* beetles, increasing until the fifth week, and beginning to decrease at the ninth week. The minor compound was produced steadily without significant change up to 9 weeks. Our results represent a step forward in the knowledge of the chemical communication of this important pest.

## 1. Introduction

Powderpost beetles are a common pest group for dried cured wood, especially the sapwood of hardwood. They convert wood into a mass of powdery or pelleted frass during the larval stage [1]. The beetles cause major destruction to wood and its products, such as hardwood flooring, hardwood timbers, plywood, crating, and furniture [2]. The damage is slow and inconspicuous; thus, infestation is generally identified belatedly due to a lack of knowledge on how to locate and monitor them. Powderpost beetles occur worldwide, having indigenous species plus introduced species in each region, but are mainly found in tropical and arid areas [3,4].

*Lyctus africanus* is a powderpost beetle belonging to the subfamily Lyctinae in the family Bostrichidae [1,5]. The species is endemic to East Asian and East African regions [6,7]. This beetle is recognized as an economically important *Lyctus* species due to the recent explosions of infestation in some areas [8]. The beetle is reported to infest dried roots, seeds, and tubers [6,9] as well as timber and timber products [6,7].

Chemical signaling is a primary strategy in many insect species for communication among individuals. A semiochemical is a chemical signal released from one organism that generates a behavioral or physiological response in other members of the same or different species. Knowing the specific semiochemical of an insect is considered a promising component in integrated pest management (IPM) programs for controlling insects. Different control strategies, such as monitoring, mass trapping, mating disruption, and attract-and-kill and push-pull strategies have been developed using semiochemicals [10]. An aggregation pheromone is one type of semiochemical. Released by an individual, it generates aggregative behavior in members of both sexes or of the same sex as the sender [11,12]. Many non-social Arthropod species, such as beetle, release an aggregation pheromone to attract their conspecific, and some conspicuous similarities of pheromone function are shown in different taxa [11,12,13,14]. We recently began the development of a monitoring system by determining the aggregation pheromones in the *L. africanus* species [15]. Three esters (Figure 1) were identified as male-specific compounds: 2-propyl dodecanoate (compound **1**), 3-pentyl dodecanoate (compound **2**), and 3-pentyl tetradecanoate (compound **3**). There was a synergistic effect among the three synthetic esters compounds; however, compound **2** was recognized to play the main role in the aggregation behavior of *L. africanus*, especially in female beetles.

This study examined the physiological and behavioral parameters that affect pheromone production by *L. africanus* to provide useful information for pest management. Information on lifespan and the effects of aging and mating status on pheromone production in males of *L. africanus* is important to determine the optimum survival conditions for adult beetles. It has been suggested that only individuals with a good ability to survive engage in intensive signaling [16]. Most members of the Bostrichidae family are non-feeding at the adult stage. The *L. africanus* are classified into a group that can taste and feed on wood [17]; hence, optimum lifespan conditions are evaluated in starved and unstarved conditions. Further, observation of the effects of age and mating status of adult male *L. africanus* beetle is essential to understanding the source of variation in pheromone production and the optimum conditions for chemical compound production. Knowing changes in pheromone production of beetles is basic knowledge to develop further research using pheromone as a pest management tool. This is important for further studies on behavioral, experimental, and physiological factors influencing pheromone signaling, and also for developing monitoring strategies for pest management.

## 2. Materials and Methods

### 2.1. Insect Source

Adult *L. africanus* beetles were cultured in a mixture of solid wood-based artificial diet [18,19] consisting of dried yeast (24%, Asahi Food and Health Care Co., Ltd., Sumida-ku, Tokyo, Japan), starch (50%, Nacalai Tesque, Kyoto, Japan), and lauan (*Shorea* spp.) wood sawdust (26%). The beetle cultures were kept in glass jars (450 mL) in a dark climatic chamber with a temperature of 26 °C ± 2 °C and relative humidity of 65% ± 10%. These beetles have been maintained in the laboratory for at least 10 years (T. Yoshimura, personal communication) in the Deterioration Organisms Laboratory (DOL) at the Research Institute for Sustainable Humanosphere, Kyoto University, Gokasho, Uji, Kyoto, Japan. Newly emerged *L. africanus* beetles were used in this study. Adult female and male beetles were sexed and separated under a stereo microscope (Leica MZ6, Leica Microsystems GmbH, Wetzlar, Germany) by apical hairs present along the hind margin of the abdominal segment [6].

### 2.2. Life Span of Adult Male L. africanus in Starved and Unstarved Conditions

To test the optimum conditions for adult beetle survival, a lifespan experiment was conducted in starved (cultured without food) and unstarved (cultured with food) conditions. The adults of a related species, *L. brunneus*, have been reported as not feeding on wood [20]. In this study, five pairs of males and females *L. africanus* were put in an individual petri dish (Ø 5 cm, 1 cm in height, BD Biosciences, San Jose, CA, USA) and covered with Ø 5 cm filter paper (Whatman No. 2, GE Healthcare, Buckinghamshire, UK) to facilitate the movement of beetles. A small piece of air-dried wood-based solid food (2 × 2 × 1 cm, 3.3 g) was placed in the Ø 5 cm Petri dish for the unstarved condition, and no food was placed in the starved condition dish. All Petri dishes were arranged randomly in the dark climatic chamber and observed weekly for mortality. The observation was terminated when all beetles died. Fifteen replications were made for each condition.

### 2.3. Effects of Mating Status on Pheromone Production

The experiment was set up to determine whether mating history affect pheromone production. Here, we used unmated and mated male beetles for the chemical analysis of pheromone compounds. To prepare the unmated and mated male beetles, the pupal-stage *L. africanus* beetles were harvested from the artificial diets, then placed individually in a small container (Ø 5 cm, 1 cm in height, BD Biosciences, San Jose, CA, USA) until they became new adults. The adults (1–5 days old) were sexed and separated for further examination. The unmated male beetles were kept separately from female beetles until the chemical analysis. To prepare the mated beetles, males and females of newly emerged adult beetles were transferred into the Ø 5 cm Petri dish covered with a Whatman filter paper No. 2 with 2:3 male-to-female ratio. The greater number of female than male beetles was intended to ensure the opportunity of male beetles to mate. One dish contained 15 adult beetles with eight replications were made for each treatment (mating status).

### 2.4. Effects of Aging on Pheromone Production

This experiment was set up to determine the effects of time or aging on pheromone production. For one set of experiments, five pairs of newly emerged male and female adult beetles were transferred into a Ø 5 cm Petri dish covered with a Whatman filter paper No. 2. Another small piece of food as described above was placed in the dish to prolong the life span of the beetles. The Petri dishes were positioned randomly in the dark climatic chamber. The males were separated from the females by the above-mentioned method. Observation on pheromone compounds was conducted by GC-MS measurement on 1-, 3-, 5-, and 9-week-old male *L. africanus* beetles using the whole-body extraction method. Eight replications were conducted for each variation.

### 2.5. Chemical Analysis

#### 2.5.1. Collection of Chemical Compounds

To collect chemical compounds, whole-body extraction with hexane was performed for both mated and unmated male beetles (1–5 days old), using the method described in previous studies to extract pheromones from other beetles and millipede groups [21,22,23]. Briefly, each beetle was immersed in hexane (10 µL) for 5 min, and 1 µL of aliquot was injected into a gas chromatography–mass spectrometry (GC–MS) instrument. Authentic compounds **1**–**3** were prepared and synthesized in a similar manner to our previous study [15].

#### 2.5.2. GC–MS Analysis

The GC–MS analysis was conducted with a Network GC System (6890N; Agilent Technologies, Santa Clara, CA, USA) coupled with a mass selective detector (5975 Inert XL; Agilent Technologies, Santa Clara, CA, USA) operated at 70 eV. The column used was an HP-5MS capillary column (Agilent Technologies, Santa Clara, USA, 0.25-mm I.D. × 30 m, 0.25-µm film thickness). The carrier gas was helium with a constant flow rate of 1.00 mL/min. Samples were analyzed in the splitless mode with the temperature programmed to change from 60 °C (initially for 2 min) to 290 °C at a rate of 10 °C/min. The final temperature (290 °C) was then maintained for 5 min. The GC–MS data were recorded using Chemstation (Agilent Technologies, Santa Clara, CA, USA) with reference to an MS database (Agilent NIST05 mass spectral library, Agilent Technologies, Santa Clara, CA, USA).

#### 2.5.3. Quantitative Determination of Three Ester Compounds

A calibration curve was constructed for each compound. The curve was obtained by correlating the GC–MS response data of the crude extract of beetle with each concentration of three standard solutions. A synthetic sample of each ester (2-propyl dodecanoate, compound **1**; 3-pentyldodecanoate, compound **2**; and 3-pentyl tetradecanoate, compound **3**) was diluted with hexane. The following concentrations were prepared: 5, 10, and 25 ng/μL; a 200 ng/μL solution was also prepared for the major compound (2). A calibration curve was then constructed.

### 2.6. Data Analysis

To determine the lifespan of male beetles, survival rates of adult male beetles were recorded weekly defined by fraction surviving. The fraction surviving (*Li*) was calculated by dividing the number of male beetles surviving at the age *i* by the initial number of adult male beetles [24]. Then, a Kaplan–Meier analysis followed by a log-rank method with a confidence interval (CI) of 0.05 was used to determine the survival duration of the beetle for 15 weeks of observation. The mean quantities of pheromone detected over the lifespan were subjected to log transformation and one-way analysis of variance (ANOVA), followed by Tukey’s HSD as a post hoc test with CI of 0.05. Meanwhile, the means of pheromone responses by mating status effect were subjected to a Mann–Whitney U test with CI of 0.05.

## 3. Results

### 3.1. Life Span of Adult Male L. africanus in Starved and Unstarved Conditions

Figure 2 shows lifespan data of male *L. africanus* adults. Throughout their lives, the survival of starved male beetles was lower than that of unstarved beetles. The number of starved male beetles dropped precipitously with time, whereas that of the unstarved beetles decreased gradually until the end of their life period. The longest lifespan of an unstarved male beetle was 14 weeks, while that of starved male beetles was only 5 weeks. The Kaplan–Meier analysis indicated that unstarved beetles survived significantly longer than starved beetles.

### 3.2. Effects of Mating Status on Pheromone Production

Figure 3 shows the quantity of each pheromone compound measured in the mated and unmated males. The figure reveals a tendency toward higher levels of all three ester compounds in the pheromones of mated males compared to unmated ones. The difference between mated and unmated males, however, only reached significance in compound **1**; the quantities of compounds **2** and **3** were not significantly different.

### 3.3. Effects of Aging on Pheromone Production

As shown in Figure 2, the maximum lifespan of male *L. africanus* was 14 weeks, but very few beetles survived this long. Thus, pheromone measurement was not conducted at 14 weeks.

Table 1 presents the quantitative fluctuation of the pheromone compounds over time. Pheromone compounds, particularly compounds **2** and **3**, were produced in beetles throughout their lives from youth until death. There were no differences in the quantity of pheromone compound **1** by time. However, compounds **2** and **3** were both initially produced at low amounts by young *L. africanus* beetles, then increased until week 5, before subsequently decreasing at week 9.

## 4. Discussion

This study indicates that the survival of starved male *L. africanus* beetles was lower than that of beetles provided food, which means *L. africanus* might utilize the artificial diet. As mentioned previously, the adults of *L. africanus* are classified into a group that can taste and feed on wood. As reported by Cymorex and Schmidt [17], the gnawing or feeding behavior and feces production of five species of *Lyctus* were quantitatively and qualitatively analyzed using an artificial diet. The *L. africanus* were observed to gnaw, feed, and produce feces on wood materials, such as wood carvings. The feeding behavior can be related to the specialization of gut anatomy, where the development of fine villi in the adult mid-gut section of *L. africanus* has been observed. The *L. africanus* is suggested to have a more functional digestion system than other species, *L. brunneus* or *L. planicollis*. The food may not only act as nutrition and provide a hiding space or means for reducing stress but also as a water source from polysaccharide decomposition in dry wood.

In addition, the male *L. africanus* beetle showed a tasting behavior like males of a closely related species, *L. brunneus* [20]. In general, tasting behavior is performed by females of *Lyctus* beetle in order to select an oviposition site, as the eggs survive best in starchy wood. A 1984 report on other beetle species in the Bostrichidae family stated that beetles with lifespans longer than 1 month require feeding for reproduction [25]. As for *L. africanus*, the lifecycle of beetles is around 2.5 months with a suitable diet for producing a new generation after the introduction of the artificial diet [19]. As mentioned earlier, *L. africanus* adults survived longer in the unstarved condition. Based on these results, the adult *L. africanus* beetles were then maintained with the wood-based diet to prolong their lifespans.

During their life spans, the lyctines are able to mate and oviposit fertile eggs within 24 h of emergence [26]. Mating is known to trigger immediate changes in the physiology and behavior of insects. This study revealed that chemical changes in the male *L. africanus* beetle occurred after mating. The results of the analysis of chemical compounds measured on mated and unmated male *L. africanus* beetles indicate that the quantity of compound **1** was affected by the mating status of male *L. africanus* beetle, while those of compounds **2** and **3** were relatively unaffected. In other words, the production of compounds **2** and **3** of the aggregation pheromone components of the *L. africanus* beetle was unaffected by the mating history. In general, the mating history of male insect beetles had no effect on insect male-produced pheromones [27]. One study reported that the female of an ant species, *Leptothorax gredleri*, was promptly avoided by males after mating, coinciding with immediate changes of the female cuticular hydrocarbon (CHC) profile, known as a pheromone. The modification of the CHC produced by the females was detectable by males, allowing them to distinguish between mated and unmated females [28].

Compound **1** is a minor compound found in the aggregation pheromone of the male *L. africanus* beetle, suggesting it would have a minor effect on aggregation behavior [15]. There was a synergistic effect among the three synthetic compounds when they were blended, indicating the essential role of compound **2** in the aggregation behavior of *L. africanus*. The other single compounds **1** and **3** induced an insignificant effect on the beetle’s responses when used alone. Hence, the increased quantity of compound **1** after mating could be affected by some factors in relation to mating activity, such as the stage of sexual maturity and the presence of female beetles. In a study on *Tenebrio molitor* (Tenebrionidae), the presence of other females acted to accelerate sexual maturation as indicated by pheromone emission rate [29]. This mechanism could also occur in *L. africanus*, which generates sexual maturity of the male beetle by the female through a mating process. However, further study is necessary to confirm the function of the compound. Pheromone exchanges frequently occur during mating, as the act of mating appears to initiate and inhibit particular pheromone compounds [30,31,32].

Measurement of pheromone compounds **1**, **2**, and **3** of the male *L. africanus* beetle indicated a continuous production of the compounds throughout their lives. As illustrates in Table 1, compound **1** production was relatively stable over time. A quantitative fluctuation of the pheromone compounds with time was found in compounds **2** and **3**; however, both were initially produced in low amounts by young *L. africanus* beetles, then gradually increased until week 5, before subsequently decreasing at week 9. Similar patterns have been reported in other Bostrichidae family members, such as *Rhyzopertha dominica* [32]. Dominicalure-1 and Dominicalure-2 as pheromone components of *R. dominica* were released by male *R. dominica* in higher quantities when the insects were relatively young, then declined in significant numbers at 24 weeks, and remained stable thereafter. The decrease of emitted pheromone reduces the insect’s lifetime reproductive fitness since the male-produced pheromone primarily attracts females as potential mates. Hence, aging could lower the recruitment rate by the insect. The current study found compounds **2** and **3**, major components of aggregation pheromone in male *L. africanus* beetle, were affected by aging. As previously mentioned, although the unstarved male *L. africanus* beetle could survive until 14 weeks, the production of those major compounds started declining at 9 weeks. In general, aging is known as a process that impacts a broad functional decline in health, increasing vulnerability to diseases and death, and reducing reproductive output. One study reported that composition changes of CHC were found in both male and female *Drosophila melanogaster* consistently with age [33].

Table 1 reveals that compound **2** was by far the dominant aggregation pheromone component in male beetles, as reported in our previous study [15]. This compound was detected in large amounts in every measurement, followed by compound **3** and compound **1**. These results suggest that compounds **2** and **3** might have an important function in the aggregation behavior of *L. africanus*, especially in the signaling behavior for mating opportunities and oviposition. We previously found that compound **1** is a minor compound in the aggregation pheromone of *L. africanus* [15]. This study clarified the minor effect of compound **1** in the aggregation pheromone of *L. africanus*, but we were not able to elucidate its role in the aggregation behavior of *L. africanus*. Further study is required to confirm the roles of each compound and their combination in the aggregation pheromone of *L. africanus*. The present information will help foster a better understanding of the chemical communication of the insect, which may be important for the development of improved pheromone-based management strategies for controlling *Lyctus* beetles.

## 5. Conclusions

This study indicated that the adult *L. africanus* survived longer in an unstarved condition. The food may act not only as nutrition but also as a hiding space or means for reducing stress. It was suggested that aggregation pheromones produced by male *L. africanus* beetle, which consists of two major compounds **2** and **3,** were affected by age, with maximum production at the 5th week, and not by the mating status. However, the other minor compound **1** was produced steadily without significant change up to 9 weeks, and its production was affected by the mating status of the male *L. africanus* beetle.

## Figures and Tables

**Figure 1 insects-12-00008-f001:**
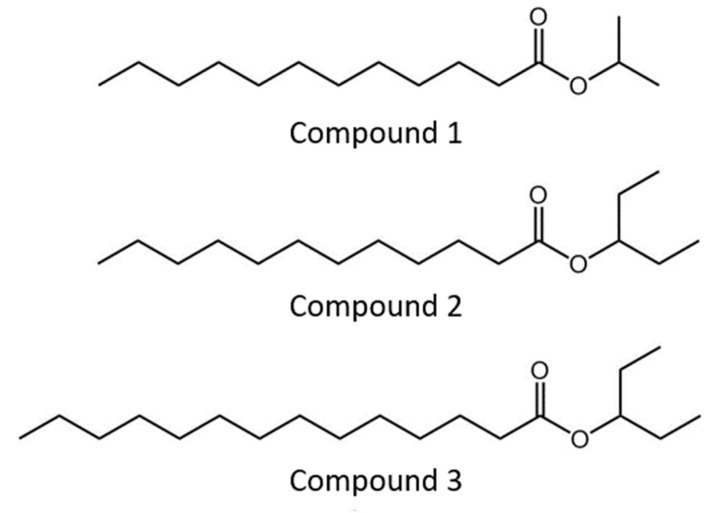
Chemical structure of ester in aggregation pheromone of *Lyctus africanus*.

**Figure 2 insects-12-00008-f002:**
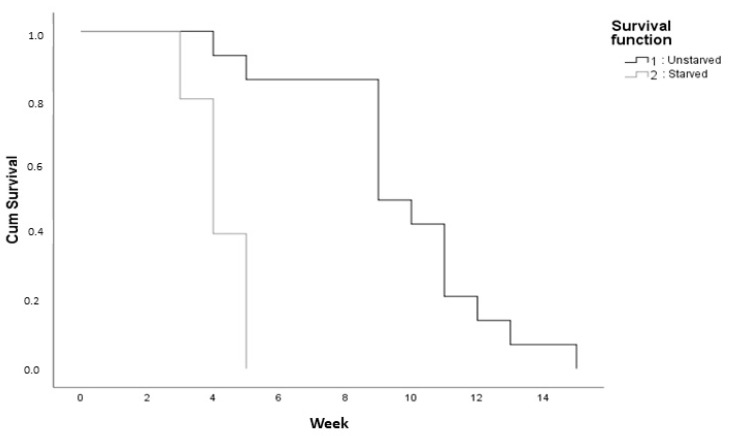
Survival function of starved and unstarved males of *L. africanus*. (Kaplan–Meier analysis, *p* = 0.00).

**Figure 3 insects-12-00008-f003:**
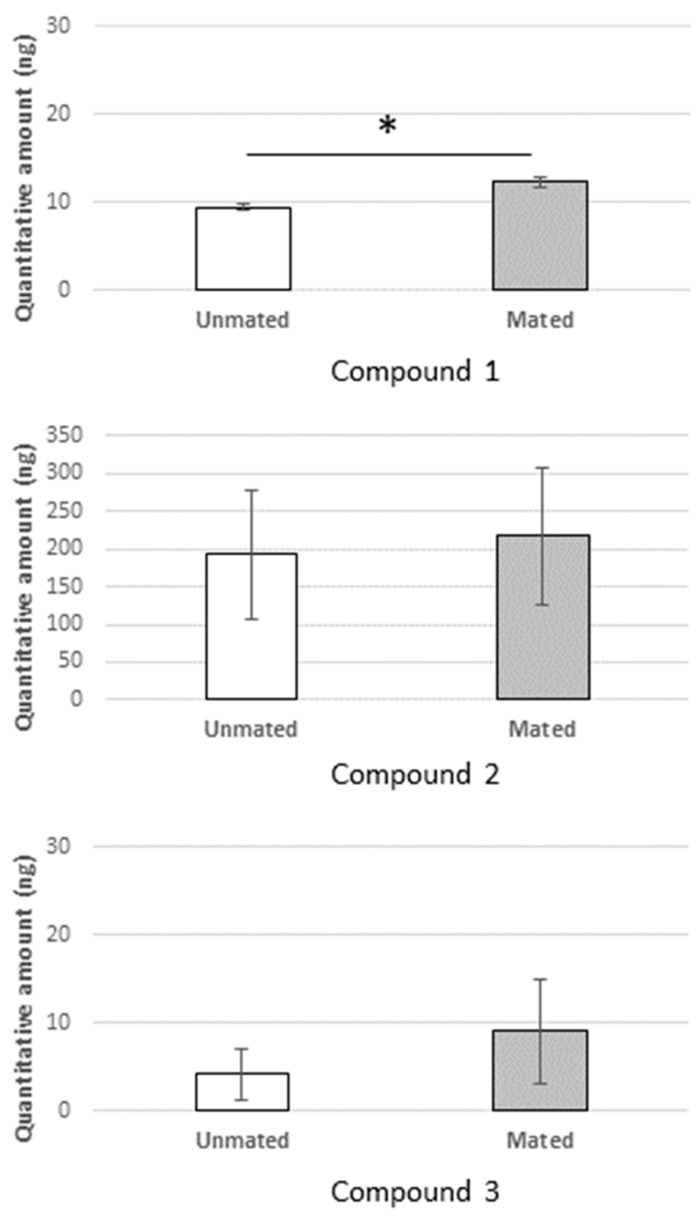
Pheromone production (compound **1**, **2**, and **3**) in the mated and unmated adult male *L. africanus*. Error bar represents the standard deviation, asterisk indicates a significant difference (*p* < 0.05).

**Table 1 insects-12-00008-t001:** Mean production of pheromone compounds by *L. africanus* at different ages.

Time (Weeks)	Pheromone Quantity (ng)		
	Compound 1	Compound 2	Compound 3
1	0.00 ± 0.00 ^a^	93.14 ± 19.15 ^b^	2.22 ± 1.93 ^c^
3	8.06 ± 2.31 ^a,^*	142.73 ± 45.31 ^b^	14.84 ± 6.87 ^b,c^
5	21.67 ± 15.04 ^a^	1185.15 ± 412.60 ^a^	186.73 ± 56.36 ^a^
9	5.84 ± 3.60 ^a^	376.34 ± 154.50 ^a,b^	65.07 ± 20.31 ^a,b^

Note: * Same letter within a column indicates no significant difference (Tukey-Kramer HSD test; *p* < 0.05) following one-way ANOVA.

## Data Availability

Data set available on request to corresponding author.

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
