# Peer review of "Influence of Age and Mating Status on Pheromone Production in a Powderpost Beetle Lyctus africanus (Coleoptera: Lyctinae)"

_insects, 2020, doi:10.3390/insects12010008_

Round 1

Reviewer 1 Report

This paper looks at the survival of fed and starved males of the powderpost beetle Lyctus africanus, and the effect of both male age and mating status on the aggregation pheromone production and composition.

The study seems to be sensibly designed, however I have some concerns over the sample size used for the pheromone analysis, and the way the pheromones were analysed (see below).

The writing was clear overall, although there were some small mistakes throughout and the abstract in particular could be improved. I think the authors could also strengthen this manuscript by making the implications of their findings clearer, and putting them in a more general context. At present it is not clear to people that don’t study this particular species why this study was necessary and interesting.

I give more detailed comments below:

Abstract and summary were actually written less clearly than the rest of the manuscript (to me at least). From the abstract neither the motivation behind the study, or the implications of the findings were obvious to me.

Introduction – the authors mention that knowing changes in pheromone production in males will be useful for pest control – but do not explicitly say why. I think this should be expanded and made clearer to make the paper more suitable for a general audience.

Figure 3 – I would label these graphs as compound 1, 2 and 3. Having just the numbers confused me at first.

L106 onwards – the authors say males have been reported to not feed on wood- but the food provided is wood based? Do you mean they might not feed on wood and that is what you are testing? Also the wording should be changed so it is clearer what this “food” actually is.

When you say eight replicates were done, do you mean 8 males were extracted and analysed in total for each experiment? If so that seems a bit low. Or was more than one male included per replicate?

Did you include an internal standard when extracting the pheromones?

I’m more familiar with mating pheromones rather than aggregation so perhaps this does not apply, but in insect mating pheromones often the ratio of the compounds matters more than the total amounts. Did you analyse the compounds are ratios or relative to the total pheromone amount? If so did your conclusions remain the same? To me it seems the relative ratios may change quite drastically over the males’ lifespan.

L246-248 I think this was meant to be deleted.

Only in the discussion did I really understand why the authors tested adult feeding – perhaps some of this could go in the introduction? Again, what exactly is this artificial diet?

In the discussion, as in the introduction, more time needs to be spent on why this information is useful – artificial lures, disruption etc etc. It might also be interesting to consider if your results suggest any costs to pheromone production or not.

Reviewer 2 Report

I reviewed the manuscript "Influence of age and mating status on pheromone production in a powder post beetle Lyctus africanus Lesne (Coleoptera: Lyctinae).  The title is very appealing and I agree that it is important to understand if different physiological status of the insect are inducing effects on the production of pheromones in this species. Moreover, I realized that the authors (at least some of them) have a long documented experience on the target pest and on the  the biochemistry of the pheromone.

However, there are some criticisms in the manuscript:

  1. some of the experiments are not well described. I am suggesting to improve the details of each experiment;
  2.  the first experiment (life spam on fed and starving beetles) is not very clear: in the first sentence, the authors are reporting that "to test the effect of age on pheromone production" they set up this bioassay having the food as a variable. Moreover, did they start with newly emerged adults? In such  so simple experiment, why the authors did not evaluate other important parameters, such as female fertility and if there was a difference in longevity and amount of feeding among different genders in the two experimental conditions?
  3. the second (effects of mating) and third (effects of age)experiments have both just 8 replicates. watching at the figure 3 and table 2, my feeling is that with a more suitable number of replicates, it would be possible to have a stronger data set. For instance, in the table, the "compound 1" has no records at the third week: why? At least, the authors should explain the missing data.
